# The Impact of a Culinary Coaching Telemedicine Program on Home Cooking and Emotional Well-Being during the COVID-19 Pandemic

**DOI:** 10.3390/nu13072311

**Published:** 2021-07-05

**Authors:** Julie K. Silver, Adi Finkelstein, Kaya Minezaki, Kimberly Parks, Maggi A. Budd, Monique Tello, Sabrina Paganoni, Amir Tirosh, Rani Polak

**Affiliations:** 1Department of Physical Medicine and Rehabilitation, Massachusetts General Hospital, Harvard Medical School, Boston, MA 02129, USA; julie_silver@hms.harvard.edu (J.K.S.); spaganoni@mgh.harvard.edu (S.P.); 2Department of Physical Medicine and Rehabilitation, Brigham and Women’s Hospital, Harvard Medical School, Boston, MA 02129, USA; 3Department of Physical Medicine and Rehabilitation, Spaulding Rehabilitation Hospital, Harvard Medical School, Boston, MA 02129, USA; 4Department of Nursing, Faculty of Life and Health Sciences, Jerusalem College of Technology, Jerusalem 95483, Israel; adilan@netvision.net.il; 5Sackler School of Medicine, Tel Aviv University, Tel Aviv 6997801, Israel; kayanuhea13@gmail.com; 6Department of Cardiology, Massachusetts General Hospital, Harvard Medical School, Boston, MA 02462, USA; KAPARKS@mgh.harvard.edu; 7Department of Cardiology, Newton Wellesley Hospital, Harvard Medical School, Boston, MA 02462, USA; 8Department of Spinal Cord Medicine, VA Boston Healthcare System, Harvard Medical School, Boston, MA 02130, USA; maggibudd@gmail.com; 9Department of Psychiatry, Harvard Medical School, Boston, MA 02130, USA; 10Department of Internal Medicine, Massachusetts General Hospital, Harvard Medical School, Boston, MA 02114, USA; MTELLO@mgh.harvard.edu; 11Division of Endocrinology, Diabetes and Metabolism, Sheba Medical Center, Tel-Hashomer and Sackler School of Medicine, Tel-Aviv University, Tel Aviv 5262000, Israel; Amir.tirosh@sheba.health.gov.il; 12Sheba Center of Lifestyle Medicine, Sheba Medical Center, Tel-Hashomer and Sackler School of Medicine, Tel-Aviv University, Tel Aviv 5262000, Israel

**Keywords:** telemedicine, telehealth, virtual visit, rehabilitation, culinary medicine, culinary coaching, COVID-19, pandemic, coronavirus

## Abstract

The coronavirus pandemic enforced social restrictions with abrupt impacts on mental health and changes to health behaviors. From a randomized clinical trial, we assessed the impact of culinary education on home cooking practices, coping strategies and resiliency during the first wave of the COVID-19 pandemic (March/April 2020). Participants (*n* = 28) were aged 25–70 years with a BMI of 27.5–35 kg/m^2^. The intervention consisted of 12 weekly 30-min one-on-one telemedicine culinary coaching sessions. Coping strategies were assessed through the Brief Coping with Problems Experienced Inventory, and resiliency using the Brief Resilient Coping Scale. Home cooking practices were assessed through qualitative analysis. The average use of self-care as a coping strategy by the intervention group was 6.14 (1.66), compared to the control with 4.64 (1.69); *p* = 0.03. While more intervention participants had high (*n* = 5) and medium (*n* = 8) resiliency compared to controls (*n* = 4, *n* = 6, respectively), this difference was not significant (*p* = 0.33). Intervention participants reported using home cooking skills such as meal planning and time saving techniques during the pandemic. The key findings were that culinary coaching via telemedicine may be an effective intervention for teaching home cooking skills and promoting the use of self-care as a coping strategy during times of stress, including the COVID-19 pandemic.

## 1. Introduction

The novel coronavirus (COVID-19) pandemic that began in late 2019 and quickly spread throughout the world in 2020 had profound effects on many aspects of healthcare delivery. For example, there was a marked increase in the use of virtual visits which is often referred to as telemedicine and has been defined as “the remote delivery of healthcare services and clinical information using telecommunications technology” [1]. The use of telemedicine has been previously described in the lifestyle medicine literature [2], inclusive of culinary medicine [3,4]. Interventions delivering health coaching via telemedicine have been shown to be effective for weight loss and improved metabolic markers [5] in participants with obesity [6] as well as prediabetes [7] and diabetes [8]. While telemedicine had previously been established as effective for healthy lifestyle changes and associated with positive clinical outcomes [9], the pandemic triggered the regulatory changes that increased reimbursement and facilitated rapid widespread adoption [10,11,12,13,14,15].

The consequences of the pandemic, including social restrictions that encouraged many people to spend more time at home, had profound effects on population health which were mostly deleterious. On the positive side, motor vehicle accidents decreased (likely due, in part, to a decrease in commuting to work), and when sports activities were curtailed, there were fewer injuries to athletes (e.g., concussion) [16]. However, negative effects have dominated the literature and public news [16]. For example, reports from numerous countries described an increased prevalence of mental health issues such as anxiety, depression and insomnia [16,17,18,19], as well as challenges in the management of chronic diseases [20,21], which contributed to a decrease in the self-rated health of people with chronic illness [22].

The social restrictions also had a profound effect on health behaviors. A report from Italy described both an increase and decrease in adherence to healthy diets [18], and reports from the United States (US) described both increases and decreases in physical activity levels, some of which may be correlated with anxiety [17]. In Poland, there were descriptions of both weight loss and weight gain [23] which might result in an increased risk of morbidity and mortality from COVID-19 [24]. 

In this study, we aimed to assess the home cooking practices, emotional well-being, coping strategies and resiliency of participants in an ongoing randomized controlled trial (RCT) during the COVID-19 pandemic that focused on improving nutrition and weight through a home cooking telemedicine intervention. 

## 2. Materials and Methods

This culinary coaching telemedicine study is a bi-center RCT aimed at evaluating the impact of a remote home cooking intervention on the nutrition and weight of participants with overweight or obesity. Due to the events surrounding the pandemic, we expanded our study to evaluate the impact on current participants’ home cooking and resiliency and the potential use of home cooking as a coping strategy. This study was approved by both sites’ institutional review board: protocol #2018P002115 (NCT03823469). This study conformed to STROBE guidelines.

### 2.1. Setting and Participants

Participants were recruited at two sites, Spaulding Rehabilitation Hospital (Spaulding), Massachusetts, US, and Sheba Medical Center (Sheba), Israel. Inclusion criteria included: body mass index (BMI) equal to or greater than 27.5 kg/m^2^ and equal to or lower than 35 kg/m^2^; primary food provider of the household who consumes fewer than five home cooked lunches and dinners per week; age 25–70 years; stable oral medications for the past 3 months; well controlled type 2 diabetes (Hga1C < 7.5; stable oral medications for the past 6 months). Exclusion criteria included pregnancy, breastfeeding, 5% < weight loss and taking weight loss medications or medications likely to cause weight gain within the past 6 months. Participants were randomly assigned, using simple randomization analogous to flipping a coin, at each site to either an intervention or control group. Both groups completed a 30-min nutritional counseling session at baseline and after three months. The intervention group completed a 3-month culinary coaching telemedicine program (CCTP), while the control group was provided with access to nutritional education resources (e.g., websites). 

### 2.2. The Culinary Coaching Telemedicine Program

A detailed description about the CCTP has been previously published [3,25]. Briefly, this program included 12 weekly, one-on-one, 30-min tele-sessions through a virtual Zoom^®^ platform that allows for synchronous face-to-face contact. At the first session, participants identified their vision regarding home cooking and 3-month goals. During each subsequent session, participants reviewed their progress toward reaching the prior week’s culinary goals and identified goals for the coming week using a self-discovery process facilitated by the culinary coach. When participants identified a new culinary skill that was necessary for their progress, they were either instructed by the culinary coach through discussions or referred to specific program resources (e.g., recipe, video). 

### 2.3. Pandemic Evaluation 

This evaluation was designed to assess the impact of the CCTP on participants’ home cooking and related coping and resiliency during the pandemic. Three questionnaires were distributed by email with up to three reminders during the first wave of the COVID-19 pandemic (April–May 2020). Participants (*n* = 38) who were enrolled in the RCT by May 2020 were included. This included 31 participants at Spaulding (13 intervention, 18 control) and 7 at Sheba (4 intervention, 3 control). Two questionnaires were validated instruments—Brief Coping with Problems Experienced (COPE) Inventory [26,27] and Brief Resilient Coping Scale (BRCS) [28] (see Appendix B for details). To best address the impact of the home cooking intervention on participant coping strategies, an additional self-care coping strategy subscale was developed and embedded in the COPE. The third questionnaire, hereafter called qualitative assessment (QA), included 5 open-ended questions to assess the impact of the pandemic on home cooking, and the use of culinary coaching strategies to address the pandemic-related stress and social isolation. 

### 2.4. Data Analysis

Subjects’ demographics are presented using descriptive statistics. Included are tests of whether these differ between the control and intervention groups. Quantitative variables were tested with t-tests and qualitative variables with chi-squared tests. Outcomes were analyzed with inferential tests using two-sample Wilcoxon rank-sum statistics. Where a variable was completely categorical, chi-squared tests were used (e.g., overall resiliency levels). Due to the small sample size and the fact that data for potential confounders were collected before the pandemic, as well as the goal being primarily descriptive, only univariate analyses are presented (i.e., only one predictor at a time was used so that each analysis had two variables: the outcome and one predictor/independent variable).

### 2.5. Qualitative Analysis

Inductive thematic analysis was used following Braun and Clarke’s guidelines [29]: Open answers were read and re-read until a list of coded extracts of data was identified and emergent themes were noted. Next, the themes were reviewed through re-reading, sifting and refinement. Initial themes were expanded or merged, and discrepant data were sought until generating clear definitions and naming each theme. Compelling extract examples were selected. All steps in the analysis process were discussed with team members until consensus was reached. 

## 3. Results

Of the 38 eligible participants, 28 (73.7%) completed the additional evaluation—including 22 at Spaulding (22/31; 71%) and 6 at Sheba (6/7; 86%). Of the 28 total participants, 14 were from the control group and 14 were from the intervention group. Notably, three participants of the intervention group completed the instruments before starting the CCTP, five during the CCTP and six completed the instruments up to 32 weeks after completing the CCTP. Table 1 provides demographic data and a comparison between the intervention and control groups. No significant differences were noted between the groups, and both groups primarily consisted of married and employed female adults below the age of 65. There were 10 non-responders, 2 from the intervention group (both female) and 8 from the control group, with 5 females and 3 males. The mean age (SD) of the intervention group non-responders is 47 (9.9) years, and for the controls, it is 44.75 (15.9) years; the mean age for all non-responders is 43.2 (14.8) years.

### 3.1. Coping Strategies during the Pandemic

The extent to which participants utilized various coping strategies during the first wave of the pandemic is presented in Table 2. The coping strategy that participants in both the intervention and control groups used the most during the pandemic was acceptance, and the least used was behavioral disengagement. Analysis of differences between groups shows that self-care, which is the only coping strategy that was specifically addressed by the CCTP, is the only strategy that was used to a greater extent by the intervention group compared to the control group (*p* = 0.03), and it was the second most used strategy among the intervention group. A trend of using approach strategies that are explicitly addressed with health coaching was also noted. These results remain significant after excluding the three participants in the intervention group who completed the instruments before starting the CCTP (*p* = 0.02). 

The Mann–Whitney (M_W) score statistic returned interesting information; in particular, it tells us that if we randomly chose someone from the control group, then, for the first line in Table 2, the probability that they would be in a “higher” group (more “positive reframing”) than a randomly chosen member of the intervention group is 0.54 (and 1 − 0.54 = 0.46 if we want to look at it the other way: a randomly chosen member of the intervention group has a probability equal to 0.46 of a higher positive reframing score than a randomly chosen member of the control group). This statistic is also interesting because it is exactly the same as the area under the receiver operating characteristic (ROC) curve if we had estimated a logistic regression with group as the outcome/dependent variable [30,31,32].

Table 3 presents the overall resiliency levels of participants during the pandemic. The high and medium levels included more participants from the intervention group than from the control group (35.8%, 57.1% vs. 28.6%, 42.8%, respectively), and the low level of resiliency included more participants from the control than the intervention group (28.6% vs. 7.1%). While the intervention group was equipped with additional coping tools, the difference in the overall resiliency was not significant (*p* = 0.33). 

### 3.2. Pandemic-Related Changes in Home Cooking and Emotional Well-Being

Participants from both groups described similar home cooking and emotional experiences impacted by the pandemic. Participants in each group reported increased engagement in home cooking. Facilitators were time, “I’ve had more time to cook and focus on my meals” (SP04-i) (i—intervention, c—control); unavailability of restaurants and bars, “I am cooking 100% at home since I do not like takeaway and restaurants are closed” (SP28-i); and a decrease in the daily demands, “eliminating the stresses of commuting has made the shift to more frequently cooking at home easier” (SP26-i). Furthermore, participants described exploring new recipes, “…I have been trying new recipes and attempt to try 3 or 4 per week” (SP30B-c); and changes in eating habits such as eating more family meals, and in their general eating frequency, “I snacked more because I was home and it was available, and I was reading or on the computer more” (SP31-i). Changes in ingredient accessibility were also noted, including some participants who remarked on food shortages in the supermarkets, and others mentioned accessibility limitations, “I’m not going to the grocery store, so it’s harder to choose and have fresh food on the whole month” (SP24-c). 

Two participants in the control group reported they did not have the time and energy to increase home cooking. Among the reasons were the amount of time required to feed the entire family who were now at home, “Because I worked many hours, it did not leave me much time for healthy and invested cooking … [I then] had to start cooking for four people, 7 days a week, and to take care of 3 meals a day. Additionally, all that while I work from home full time” (SMC9-c); and an increased workload during the pandemic, “[I was] not paying attention to what I am eating, [or] not eating as healthy as before coming from work more tired, didn’t have the energy for eating healthier because it’s a lot of work” (SP05-c). 

Participants from both intervention and control groups described similar emotional experiences impacted by the pandemic. Participants in both groups reported on depression and stress, “In general—more eating, perhaps as a way to reduce stress” (SMC2-c); “Less motivated than normal since pandemic. Probably some depression” (SMC9-c). Participants in both groups also reported less motivation to maintain healthy eating habits, “My eating behaviors have probably gotten worse during the pandemic. [I am] less motivated than normal since the pandemic” (SP08-i). Lastly, members from both groups reported a decrease in social gatherings, “I no longer eat in bars with friends… Workplace pizza Fridays are no more” (SP17-i). 

### 3.3. CCPT-Related Changes in Home Cooking and Emotional Well-Being

Intervention group participants, even those who completed the program more than 6 months prior to this evaluation, named culinary tools and techniques that they acquired in the program and practiced during the pandemic. These included meal planning, “being more organized with meal prep, having a routine for healthy meals that I am able to cook” (SP08-i; 28 weeks from last culinary coaching session); and time saving techniques, “I use my freezer for leftovers…the skills I baselined during the coaching sessions are now easy to perform, and it takes less time to make a balanced meal from scratch than it does to cook a frozen pizza” (SP18-i; 12 weeks from last culinary coaching session). Efficient shopping was observed, “I plan my meals and trips to the grocery ahead of time to minimize exposure and go shopping once every 10 days or so” (SP18-i; 12 weeks from last culinary coaching session). Participants also utilized behavioral skills that were significant to them while dealing with the pandemic such as accountability and self-determination goals: “The program helped me to identify behaviors I would like to improve, this helped me to set goals and to accomplish them” (SMC5-i; 1 week from last culinary coaching session); “I have not let myself fall off the wagon and eat terribly” (SP17-i; 14 weeks from last culinary coaching session). The positive impact of the intervention on participants’ emotional well-being was also notable up to several weeks after the program. Participants reported that home cooking helped them to improve their sense of control, “I have focused on cooking as best as I can. It is the one control that is left in this crazy situation” (SP17-i; 14 weeks from last culinary coaching session); and to improve resiliency and hope, “Maintaining nutrition and balance during this time was very pleasing to me, especially when I heard in the media about difficulties, and I really did not feel part of the overeating problem during this time” (SMC5-i; two weeks from last culinary coaching session); “Even though I am scared about the pandemic and its possible impact on me personally and my closest relations and friends, and the economy, it has been good to also have a positive change occurring” (SP26-i; completed nine culinary coaching sessions).

Participants from the control group who met with a registered dietician twice during the study also reported healthy changes. For example, some stated they were consuming healthier food items: “It was very helpful as a general guidance, trying to improve my consumption of certain foods, as olive oil, nuts and fish” (SP24-c); “I have tried to tie in a lot of food groups when I snack (veggies, fruits, dairy and grains)” (SP30B-c). Of note, control group narratives did not have statements of improved emotional well-being. 

### 3.4. Overall Change in Nutrition, Weight and Health

Participants in the intervention group reported cooking healthy food with enhanced attention during the pandemic, “I’m eating all my meals at home so I’ve been more conscious about eating healthy meals like produce or having healthy snacks” (SP04-i). Participants primarily from the intervention group, but also from the control group, reported losing weight during the first wave of the pandemic: “I keep thinking about what I will eat before my meals. I am losing weight, and this helps me cope and continue” (SP20-c); “I’ve been losing weight and seeing real results from increasing my home meals, so from that perspective isolation has been good” (SP18-i); “being home all day could have led to weight gain but this [culinary coaching program] makes me accountable” (SP32A-i). 

## 4. Discussion

In this study, we assessed the impact of an ongoing culinary coaching RCT on home cooking practices, coping strategies and resiliency during the first wave of the COVID-19 pandemic. We hypothesized that the only coping strategy that the intervention group would use more than control is home cooking and self-care, and therefore they would be more resilient to pandemic-related stress. Our quantitative evaluation found that the intervention group used self-care as a coping strategy significantly more than the control group. While the intervention group was more resilient than the control group, this difference was not significant. In addition, the qualitative analysis provides an in-depth insight on specific tools that were used to improve nutrition up to seven months after the program completion. While nutritional benefits of home cooking interventions have been described previously, the key findings from this study suggest an additional emotional benefit to these programs during the pandemic. 

Recent studies describe pandemic-related mental health symptoms in the general population [16,33] and a need for enhanced coping, particularly for people who are overweight or obese [34]. Our study participants who were overweight or obese expressed an experience of increased stress during the pandemic. Acceptance was the coping strategy most used by our study participants in both intervention and control groups. The intervention participants also used self-care and home cooking, and they comparatively demonstrated higher resiliency (though the difference in resiliency was not significant). Acceptance is often considered a predictive or protective factor [35] in models of resiliency, and our findings support acceptance as potentially relevant to an individual’s resiliency, as acceptance was common for both groups. There have been calls to include behavioral interventions to help in coping with pandemic-related distress [16]. Further research is needed to evaluate whether investing in lifestyle medicine interventions might also yield improved resiliency. 

The interest in home cooking as a potential health behavior to improve nutrition is emerging [36,37]. In this study, participants in both the intervention and control groups reported changes in their eating habits. While both groups reported an increase in home cooking and exploring new food items and recipes, only the intervention group reported eating healthier. This suggests that the CCTP likely affected the intervention cohort’s response during the pandemic, with more opportunities to focus on nutritional changes and adopt healthier habits. Further study is needed to evaluate the specific impacts of home cooking interventions during this important time. 

Many people across the world experienced increased stress during the pandemic, and when not effectively managed, stress can affect eating [38] and health-related behaviors [29]. A recent meta-analysis showed a significant association of stress with unhealthy dietary patterns [39]. Specifically, an inverse pattern of stress and healthy dietary behaviors was observed; more stress was associated with less healthy dietary behaviors. The intervention program used in this study utilizes an individualized approach to culinary behaviors, incorporating empirically based strategies such as mindfulness [40] and motivational interviewing. Therefore, the nature of the intervention provided may explain why those in the control group did not similarly endorse increased well-being or mindfulness of self-care aspects while experiencing stressors from the pandemic that were demonstrated in the intervention group.

The pandemic also affected the way in which healthcare is delivered, particularly in regions where hospitals were inundated with infected patients who were critically ill [41]. Early in the pandemic, the rapid spread of the virus triggered urgent limitations on in-person patient visits and resulted in some regions experiencing rapid expansion of virtual visits [11,42]. To facilitate the latter, in the US and some other countries including Israel [43], there were large-scale reimbursement shifts allowing for quicker and more widespread adoption of telemedicine delivery [11,15]. Notably, even prior to the existence of COVID-19, more than half of US hospitals were using telemedicine in some form—mostly for mental health, chronic disease management and primary care delivery [44]. Payors were already shifting from volume- to value-based care models, which organically favor cost-effective and outcome-driven methods of care delivery such as telemedicine [45]. The global market for telemedicine was already projected to see substantial growth [45]; the pandemic forced explosive growth. As telemedicine was established as effective for healthy lifestyle change and improved clinical outcomes [8], as well as being strongly associated with very high patient satisfaction ratings [46] and significant cost-effectiveness [44], it is likely here to stay. 

This study adds to the literature as there is little information about home cooking interventions during the pandemic, and the findings may help to inform clinical dietary interventions now as well as in the future. Strengths of this study include the binational cohort, RCT and the opportunity to study culinary health behaviors during a unique time in history. Limitations include the lack of baseline data, and the small sample size that limits the ability of the results to be extrapolated to large populations. Thus, further studies are needed. 

## 5. Conclusions

In this study, we found that a culinary coaching intervention delivered via telemedicine might result in the adoption of specific home cooking practices and the use of self-care as a coping strategy during the particularly stressful period of the first wave of the COVID-19 pandemic. More research is needed to explore whether stressful events such as the pandemic provide an opportunity for positive lifestyle interventions.

## Figures and Tables

**Table 1 nutrients-13-02311-t001:** Demographics of participants in the culinary coaching study.

	Control (*n* = 14)	Intervention (*n* = 14)	Total (*n* = 28)	*p*-Value
Mean age, years (median, interquartile range)	51.6 (54.5, 14)	45(47, 23)	48.3 (51.5, 22.5)	0.26
Age ≥ 65, *n*	1 (7%)	0	1 (4%)	0.31
Female	10 (71%)	9 (64%)	19 (68%)	0.69
Marital Status				0.86
Married	7 (50%)	6 (43%)	13 (46%)	
Living together	2 (14%)	3 (21%)	5 (18%)	
Never married	3 (21%)	4 (29%)	7 (25%)	
Divorced	2 (14%)	1 (7%)	3 (11%)	
Employment Status				0.59
Employed	13 (93%)	12 (86%)	25 (89%)	
Student	0 (0%)	1 (7%)	1 (4%)	
Other	1 (7%)	1 (7%)	2 (7%)	
Yearly Household Income				0.73
Below the average	4 (29%)	4 (29%)	8 (29%)	
Around the average	3 (21%)	1 (7%)	4 (14%)	
Above the average	6 (43%)	8 (57%)	14 (50%)	
Unknown	1 (7%)	1 (7%)	2 (7%)	
Health Status				
Hypertension	2 (14%)	1 (7%)	3 (11%)	0.54
Hyperlipidemia	2 (14%)	3 (21%)	5 (18%)	0.62
Type 2 diabetes	0 (0%)	1 (7%)	1 (4%)	0.31

**Table 2 nutrients-13-02311-t002:** Participant coping strategies during the COVID-19 pandemic.

	Control	Intervention	*p*-Value	M_W Score
	*n* = 14, Mean (SD)	*n* = 14, Mean (SD)		
Approach strategies				
Positive reframing	5.71 (1.44)	5.43 (1.60)	0.71	0.46
Active coping	5.79 (1.12)	5.93 (1.54)	0.63	0.55
Planning	5.36 (1.55)	5.71 (1.14)	0.55	0.57
Emotional support	4.71 (1.73)	5.14 (1.75)	0.55	0.57
Use of informational support	3.86 (1.99)	4.50 (2.14)	0.31	0.61
Acceptance	6.43 (1.16)	6.57 (1.55)	0.67	0.55
Self-care	4.64 (1.69)	6.14 (1.66)	0.03	0.74
Overall approach strategies	36.5 (6.28)	39.43 (7.17)	0.26	
Avoidance strategies				
Self-distraction	5.64 (1.45)	5.57 (1.65)	0.83	0.53
Denial	2.64 (0.84)	2.42 (0.85)	0.26	0.39
Substance use	2.29 (0.83)	2.43 (1.16)	0.87	0.51
Behavioral disengagement	2.14 (0.36)	2.21 (0.43)	>0.90	0.54
Venting	4.29 (1.68)	4.07 (1.59)	0.79	0.47
Self-blame	2.79 (0.70)	3.14 (1.41)	0.77	0.52
Overall avoidance strategies	19.79 (3.68)	19.86 (4.28)	0.96	
Neither approach nor avoidance				
Humor	4.14 (1.88)	4.29 (1.90)	0.91	0.51
Religion	3.93 (1.81)	3.64 (2.34)	0.39	0.41

Two Likert scale items per strategy (1 = I haven’t been doing this at all, 4 = I have been doing this a lot).

**Table 3 nutrients-13-02311-t003:** Participant resiliency level during the COVID-19 pandemic.

Resiliency Level	Control, *n* (%)	Intervention, *n* (%)	Total, *n* (%)
Low	4 (28.6%)	1(7.1%)	5 (17.8%)
Medium	6 (42.8%)	8 (57.1%)	14 (50%)
High	4 (28.6%)	5 (35.8%)	9 (32.2%)
Total	14 (100%)	14 (100%)	28 (100%)

Number of participants in each resiliency level (low = 4–13, medium = 14–16, high = 17–20), *p* = 0.33.

## Data Availability

Data supporting the reported results can be obtained by contacting the corresponding author.

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
