# Peer review of "The Impact of a Culinary Coaching Telemedicine Program on Home Cooking and Emotional Well-Being during the COVID-19 Pandemic"

_nutrients, 2021, doi:10.3390/nu13072311_

Round 1

Reviewer 1 Report

Please see a few minor comments in the attached pdf.

Reviewer 2 Report

The manuscript of Silver et al reports the results of the telemedicine impact of culinary coaching during Covid-19.

Major revision

Introduction

The sentence from lines 75-79 seems more appropriate for discussion and conclusion, please move them.

It would be interesting in the introduction also to highlight the effect of lifestyle (eating and lifestyle habits and sports or physical activity) also on the most common chronic diseases during COVID. There are International studies in the literature that shown this aspect (Effect of COVID-19 lockdown on patients with chronic diseases, Coronavirus disease (Covid-19): How does the exercise practice in active people with type 1 diabetes change? A preliminary survey, Lifestyle habits of adults during the COVID-19 pandemic lockdown in Cyprus: evidence from a cross-sectional study; One Month into the Reinforcement of Social Distancing due to the COVID-19 Outbreak: Subjective Health, Health Behaviors, and Loneliness among People with Chronic Medical Conditions)

Methods

Could you describe better how the randomization was done?

Change the subtitle "Outcome Analysis" in "Data Analysis" and report only information on the statistical plan.

References to tables should be moved to the Results section.

Could you please rewrite better the statistical methods better? There is something unclear.

What do you mean by "bi-variable"?

Results

What are the characteristics of 10 people who lack additional evaluation? Please provide at least information on gender and age information.

Table 1 Are these the only information about participants? Nothing about the type of work or other health condition?

Table2: modify removing the M_W score and leave only the pvalue.

If you use the non-parametric test it will be better and reasonable to report median and IQR rather than mean and sd.

In lines, 169-173 might be better to report percentages rather than absolute frequencies.

Why was people's weight not reported before and after the intervention?

Discussion

Lines 296 to 304 are just a US point of view since the sample was also represented also by another country which is the situation in Israel?

Conclusion

Starting from results it may also be due to the small sample size the evidence between the two groups and from a statistical point of view was not strong enough to speak of association. Please review the sentence "suggests there maybe..."

Minor revision

Multiple typos in references (e.g. in lines 103 and 119) and in the discussion section.

Please check the BMI unit of measurement.

In table 1 change the pvalue 1 with >0.90.

Round 2

Reviewer 2 Report

Thanks the authors for their improvement in the manuscript.

Line 38 Please reviewed "4,6..." because something is missing and probably you have to change comma

Line 141 please change in "bi-variable"

Please add "years after 47 (9.9), 44.75 (15.9)  and 43.2 (14.8).

In table 1 please report >0.90 not >0.99  and the last column should be the value of the test (pvalue).

From Lines 186  please add "M_W"  after "Mann-Whitney" to have a connection with the table. At the end of the paragraph add references adequate to justify the similarity with logistic regression.

Line 350 Please change "associated"  because results, from statistical point of view, didn't show an association.
